# Long-Term Orientation and Tax Avoidance Regulations

Katarzyna Bilicka [1,2], Danjue Clancey-Shang [2,*] and Yaxuan Qi [3]

1. National Bureau of Economic Research, Center for Economic and Policy Research, Oxford University Centre for Business Taxation, Park End Street, Oxford OX1 1HP, UK; kat.bilicka@usu.edu
2. Jon M. Huntsman School of Business, Utah State University, Logan, UT 84322, USA
3. Department of Economics and Finance, City University of Hong Kong, Hong Kong, China; yaxuanqi@cityu.edu.hk
* Correspondence: danjue.shang@usu.edu

**Abstract:** In this paper, we explore the relationship between the culture of the country where a multinational corporation (MNC) is headquartered and the MNC's stock market reaction to tax avoidance regulations. Specifically, we examine the different responses of MNCs following the implementation of the 2010 UK reform that restricted profit shifting for a specific group of firms. We find that, in countries with short-term-oriented cultures, MNCs affected by this reform experienced positive stock market responses relative to their unaffected counterparts. This is not found in long-term-oriented cultures. This difference in response can partly be explained by the differing perceptions of the role tax havens play in tax minimization practices between more long-term-oriented cultures and those oriented towards the short term. We provide evidence that investors from more future-oriented cultures may recognize the short-lived effectiveness of a regulation ex ante, and thus price the quasi-exogenous market shock differently than their more short-term-oriented counterparts.

**Keywords:** long-term orientation; debt shifting; multinational companies; tax avoidance; stock market responses

## 1. Introduction

Stock markets incorporate investors' forward-looking expectations. According to the classic Intertemporal Capital Asset Pricing Model (ICAPM), asset prices are fundamentally determined by solving the representative agent's intertemporal consumption–investment problem (Merton 1973). However, the extent of the forward-looking behavior displayed by the representative agent and its consequent impact on asset prices remain empirical questions. In this paper, we aim to shed light on the answers to these questions using a cultural perspective. Specifically, we investigate whether the forward-looking culture of a multinational corporation's (MNC) home country plays a role in determining how responsive the stock market is when pricing new information.

To gauge the degree of forward-looking culture in each country, we use a metric that is part of the set of measures developed by Geert Hofstede and his colleagues to capture various dimensions of a country's cultural characteristics (Hofstede 1984, 1991; Hofstede and Minkov 2010). The six dimensions of national cultures include power distance, individualism, uncertainty avoidance, motivation towards achievement (formerly known as masculinity), long-term orientation, and indulgence. These measures have been adopted by numerous studies that investigate the impacts of national cultures on social, economic, and political outcomes.[1] In this paper, we focus on one particular measure, a country's long-term orientation, which captures the nation's tendency to focus on the future. A long-term-oriented culture places great emphasis on persistence, while its short-term-oriented counterpart focuses on quick results. The previous literature suggests that a country's long-term orientation is associated with many corporate practices. The authors of (Kitching

et al. 2016) find that, in countries with more long-term-oriented cultures, the cost stickiness in managerial resource adjustment is less pronounced. Lievenbruck and Schmid (2014) show that companies in long-term-oriented cultures hedge less. Both Wang and Esqueda (2014) and Haq et al. (2018) document that firms borrow less in more long-term-oriented cultures. Graafland and Noorderhaven (2020) document that the joint force of economic freedom and long-term orientation in a country promotes the corporate social responsibility practices of firms in that country. A country's long-term orientation is also documented to be associated with stock market behaviors. Chang and Lin (2015) find that stock markets in countries with more long-term-oriented cultures herd less. Khan et al. (2021) show that a long-term-oriented culture has a moderating effect on investors' representativeness bias when making investment decisions. Hoang et al. (2022) find that a long-term-oriented culture had a mitigating effect on the negative reactions of the stock market during the COVID episode. Several of these findings align with the notion that a culture oriented towards the long term is inclined to prioritize the long-term performance and enduring impacts of incidents and decisions. However, research on the effect of this particular cultural dimension on the financial market's response to the implementation of regulations is limited. More specifically, we know little about whether the long-term effectiveness of a particular regulation is reflected in the financial market responses across different cultures. In this paper, we explore the role of a country's long-term orientation in mediating the effects of an anti-tax avoidance regulation on financial markets.

In 2010, Worldwide Debt Cap (WDC) was implemented in the UK to limit the amount of debt that MNCs can hold in the UK relative to their total worldwide debt. The threshold was put at 75%, creating a treated group of firms—those that had their debt levels in the UK above this limit—and a control group—those that did not. We choose the WDC to study our question of interest for two reasons. First, the quasi-experimental variation that WDC created allows us to identify the causal effects of a regulation on stock markets by comparing outcomes for treated and control groups. Second, previous works in the literature have documented the stock market impacts of the WDC on affected MNCs and the transient nature of these impacts. Specifically, applying the WDC as a quasi-natural experiment, Bilicka et al. (2022b) find that the affected MNCs experienced higher stock market returns and less frequent occasions of extremely negative returns; the MNCs with worse corporate governance qualities or with access to tax havens drive the results. These results imply that firms with substantial information asymmetry experienced significantly positive stock market responses to the WDC, supporting the idea that investors see tax avoidance as a potential tool to facilitate managerial rent extraction, and anti-tax avoidance regulations have positive effects on firm values as they curb the agency problem. However, Bilicka et al. (2022a) find that the affected MNCs responded to the WDC by adjusting their cross-country debt allocations and real operations to circumvent the reform, and it took on average three years to finish the adjustments. This is consistent with the transient nature of the stock market impacts of the WDC documented in Bilicka et al. (2022b), which shows that the difference in stock market response between the affected and unaffected MNCs only lasted for three years.

Since the positive stock market response to the WDC is short-lived for the affected MNCs, in this study, we investigate whether the long-term orientation dimension of the culture in an MNC's home country plays a role in explaining the crosssectional variation of the stock market impact of this reform. If the capital market focuses on long-term effects and investors recognize the transient nature of the effectiveness of the WDC, we would expect to see less significant positive market reactions to the implementation of this reform; potentially, the long-term adverse impact of the WDC on an MNC's cash flows may even outweigh its positive effect in curbing agency problems, and lead to a negative overall influence on firm values. On the other hand, in a market that is more short-term-oriented when investors are pricing a new piece of information, we would expect to see a stronger positive reaction to the WDC. Consistent with our hypothesis, we find that only the affected MNCs headquartered in countries with short-term-oriented cultures experienced positive

stock returns following the implementation of the WDC. In addition, affected MNCs from short-term-oriented cultures are the only ones that experienced significant reductions in the frequency of extreme negative returns. We also find that, in the context of the WDC reform, an MNC's access to tax havens may be perceived differently by a country with a long-term-oriented culture compared to a country with a culture oriented towards the short term.

Our study documents novel evidence on the effectiveness of financial regulations in connection with the national cultures of countries where MNCs are headquartered. In recent years, the issue of multinational tax avoidance has surged to the forefront of policy discussions. A widely employed tactic by multinational corporations (MNCs) to reduce their global tax liability involves strategically allocating debt across various tax jurisdictions. In response, nations worldwide have implemented various anti-tax avoidance measures aimed at mitigating this specific method of profit shifting (Carrizosa et al. 2020; Serrato 2018). The conventional perspective on corporate tax avoidance suggests that engaging in such practices can provide opportunities for managerial rent extraction, and stock markets are found to associate tax avoidance and corporate practice that facilitates tax avoidance with negative connotations (Desai and Dharmapala 2009; Desai et al. 2007; Desai and Hines 2002). Tax avoidance practices are also linked to higher crash risks in the stock market (Kim et al. 2011). An anti-tax avoidance regulation such as the WDC may mitigate the agency problem, especially for the worse-governed MNCs Bilicka et al. (2022b). On the other hand, theoretically, tax planning should enhance the shareholder wealth as it positively affects corporate cash flows. Therefore, the implementation of anti-tax avoidance regulations could potentially disrupt cash flows and subsequently have an adverse impact on the overall value of the company (Goh et al. 2016; Lim 2011; Mills 1998). We complement this literature by taking a cultural perspective. We provide evidence on how the long-term orientation dimension of a national culture affects the market reaction to the implementation of an anti-tax avoidance regulation, when the regulation pertains to cross-country markets. We document that more future-oriented cultures may recognize the short-lived effectiveness of a regulation ex ante, and price this quasi-exogenous market shock differently than their more short-term-oriented counterparts.

We also contribute to the literature that studies the association between a country's culture and its financial market dynamics. Chui et al. (2010) show that individualism is positively associated with stock market trading volume and volatility, as well as the profitability of the momentum trading strategy. Dang et al. (2019) document robust evidence that firms in more individualistic cultures exhibit higher levels of crash risk, possibly caused by earnings management, excessive managerial risk-taking, and investors' difference of opinion and overconfidence. Ashraf (2021) and Hoang et al. (2022) both study the COVID-19 episode and find that the uncertainty avoidance and long-term orientation dimensions of a culture have moderating effects on stock markets' reactions to negative information. In this study, we investigate how a culture's long-term orientation manifests in financial markets in the context of an anti-tax avoidance regulation, previously documented for its short-lived effectiveness.

The rest of this paper is structured as follows. Section 2 discusses data and our empirical strategies. Section 3 provides a discussion of the empirical results. Section 4 concludes.

## 2. Data and Empirical Strategy

### 2.1. Sample Construction

We construct our data sample following Bilicka et al. (2022b). First, we collect financial and stock market information for all MNCs that maintained at least one subsidiary in the UK in 2010 from multiple data sources. The initial sample of MNCs is extracted using Osiris by the Bureau van Dijk (BvD). Then, we use the CDs of Osiris to extract subsidiaries of these MNCs year by year. Our analysis is restricted to subsidiaries where the MNC holds a majority ownership stake of 50% or more, thereby indicating effective control. For the

UK-based subsidiaries, we extract financial data from the FAME dataset to construct the ratios of UK debt to worldwide debt of an entire MNC. MNCs were subject to the WDC if this ratio exceeded 75% in 2010. These MNCs form our treatment group. Note that our use of subsidiary-level data is *only* for the purpose of constructing the debt ratios.

We collect consolidated financial data for the MNC groups from Osiris, and stock market data from DataStream. The list of tax havens adopted to construct the dummy for tax haven access is from Bennedsen and Zeume (2018). Combining these data sources, our sample covers financial and stock market data for each MNC at the group level. Our main analysis focuses on the period of three years prior to and post the WDC, but we collect data during the full period of 2003–2016 to establish the longer-term dynamics in the data we study as well. We obtain the culture measure from Hofstede's website[2] and merge it with the corporate financial and stock return data.

## 2.2. Variables of Interest

In this paper, we study the stock market response to the WDC in connection with the culture of an MNC's home country. Following Bilicka et al. (2022b), we construct five return measures to investigate the effect of WDC on the MNCs' stock market performance: annual buy-and-hold return (*Raw Ret*), CAPM $\alpha$ against FTSE100 (*Alpha UK*), CAPM $\alpha$ against S&P500 (*Alpha US*), annual abnormal return against FTSE100 (*Abn Ret UK*), and annual abnormal return against S&P500 (*Abn Ret US*).[3]

We also construct four measures to examine the impact of WDC on the incidence of extremely negative returns the MNCs experience. Following Kim et al. (2011), the crash risk indicator variables, *Crash UK* and *Crash US*, equal one for a firm-year if the firm experiences one or more crash weeks in the UK (US) market; otherwise, *Crash UK(US)* is equal to zero.[4] Following Chen et al. (2001), we compute the negative skewness variables, *Neg Skew UK* and *Neg Skew US*, as additional measures to assess the occurrence of extremely negative returns.[5]

Bilicka et al. (2021) document that it took the affected MNCs about three years to adjust their operations in response to the WDC; after that, their overall tax avoidance practice involving debt did not vary from prior to the regulation. Bilicka et al. (2022b) show that the stock market positively responded to this regulation over the 3-year window after the WDC, but then the affected and unaffected converged after. As the adjustments in both real operations and stock market measures were short-lived, we study whether the markets in more long-term-oriented cultures can foresee the time limit on the effectiveness of the WDC. We adopt the long-term orientation measure developed by Hofstede (1991) and amplified by Hofstede and Minkov (2010), which captures a national culture's tendency to focus on the future. We obtain this measure from Hofstede's website and merge it with the corporate financial and stock return data. We then split our sample by the median of this measure, and form the sample with MNCs headquartered in countries with longer-term-orientated cultures, and MNCs headquartered in countries with shorter-term-orientated cultures.

## 2.3. Empirical Approach

We use the difference-in-differences (DID) approach to investigate the differential stock market responses of MNCs to the 2010 UK reform. MNCs that were subject to the reform, i.e., those with debt ratios above the reform threshold, are in our treated group. The unaffected MNCs, with debt ratios below the threshold, are in the control group. Our approach relies on comparing outcomes of treated MNCs to those of control group MNCs. The identifying assumption behind this method is that the evolution of stock markets would be the same for both treated and control group MNCs in the absence of the WDC. As such, we estimate the following model:

$$Y_{i,t} = \alpha + \beta Affected_i \times Post_t + \gamma X'_{i,t-1} + \eta_t + \delta_i + \epsilon_{i,t} \tag{1}$$

where $Y_{i,t}$ is one of the stock return variables; $Affected_i$ is a dummy variable that equals one if the MNC $i$ is affected by the reform; $Post_t$ is a dummy variable that equals one

from 2010 onward; $X'_{i,t-1}$ is a set of control variables, such as firm size (lagged logarithm of total assets), book-to-market ratio, stock return, and stock return volatility. $\eta_t$ is the year fixed effect, $\delta_i$ is the firm fixed effect, and $\epsilon_{it}$ is the error term. Consequently, the effect of the WDC is identified by comparing stock returns of each treated firm before and after the reform relative to stock returns of control group MNCs, i.e., estimating the difference-in-differences (DID) coefficient on $Affected_i \times Post_t$. The coefficient of interest is $\beta$.

We consider the following stock market variables: the annual buy-and-hold return (Raw Ret), $\alpha$'s estimated by regressing daily returns against S&P500 (Alpha US) and FTSE100 (Alpha UK) indices, and abnormal returns based on CAPM model in which the market portfolio is proxied by either S&P500 (Abn Ret US) or FTSE100 (Abn Ret UK) indices.

We then split the sample by the median of the Hofstede long-term orientation measure (Hofstede 1991), and conduct the aforementioned regression analysis on the sub-sample of MNCs headquartered in countries with more long-term-oriented cultures, as well as those headquartered in countries with more short-term-oriented cultures. Note that such a measure captures a national culture's tendency to focus on the future. As the stock market responses to the WDC are documented to be short-lived, we examine whether the financial markets in countries with cultures with different levels of long-term orientation recognize this time limit and respond accordingly.

We further investigate the roles that tax havens play in mediating the effects of the WDC in different cultures. As suggested by the literature, tax avoidance practices are more problematic for firms with higher levels of information asymmetry. A potential channel through which the WDC can increase firm values is to alleviate the information asymmetry and curb potential agency problems. Bilicka et al. (2022b) find that the positive stock market responses are mostly driven by worse-governed firms and firms with access to tax havens. However, a more future-focused culture could view tax havens as a potential tool for further tax avoidance practice in the future, while a more short-term-oriented culture could be inclined to focus on the potential improvement in the current information asymmetry for firms with access to tax havens. Hence, we examine the differential effects of the WDC on firm values depending on whether a firm has access to tax haven affiliates. We do that for both MNCs headquartered in countries with long-term and short-term orientation cultures. We use this analysis to draw inference into the role of a culture's long-term orientation dimension in explaining the link between the anti-tax avoidance regulation and the firm values.

As the WDC can potentially improve the information environment for the affected MNCs in the short term, we also examine the effect of the WDC on the occurrence of extremely negative returns. We use $Crash\,UK(US)$ and $NegSkew\,UK(US)$ to measure the occurrence of extremely negative returns, and investigate whether the impact of the WDC plays out differently in long-term-oriented cultures and short-term-oriented cultures.

## 3. Empirical Findings

### 3.1. Descriptive Statistics

In this subsection, we discuss the summary statistics for the variables in our empirical analysis. Table 1 provides the summary statistics for the MNCs in the two sub-samples split by the long-term orientation dimension of the countries' cultures. *BM* is the book-to-market ratio. *Volatility* is the standard deviation calculated using daily returns for each firm-year combination.

*Haven* is a dummy equal to 1 when an MNC has at least one tax haven affiliate in the firm structure where it owns more than 50% shares of that affiliate. *ROA* is the ratio of profit and loss before taxes to total assets, and *Leverage* is the ratio of long-term liability to total assets.

**Table 1.** Summary statistics: long- vs. short-oriented cultures.

| Variables | N | Mean | Sd | Min | p25 | Median | p75 | Max |
|---|---|---|---|---|---|---|---|---|
| **Panel A: MNCs from Long-Term-Oriented Culture** | | | | | | | | |
| Raw Ret | 3920 | 9.067 | 52.23 | −94.71 | −24.36 | 3.025 | 32.90 | 393.7 |
| α US | 3902 | 0.0125 | 0.187 | −0.975 | −0.0861 | 0.0109 | 0.113 | 3.225 |
| α UK | 3902 | 0.0117 | 0.183 | −0.984 | −0.0863 | 0.0141 | 0.112 | 3.378 |
| Abn Ret US | 3737 | 2.669 | 44.87 | −178.4 | −23.93 | −3.866 | 20.59 | 380.3 |
| Abn Ret UK | 3737 | 1.462 | 44.72 | −178.3 | −24.52 | −4.985 | 19.07 | 342.6 |
| Long-term orient | 3920 | 79.75 | 12.38 | 52.90 | 73.55 | 87.41 | 87.91 | 100 |
| Volatility | 3920 | 2.583 | 1.359 | 0.648 | 1.831 | 2.349 | 3.085 | 52.61 |
| Haven | 1060 | 0.673 | 0.469 | 0 | 0 | 1 | 1 | 1 |
| Roa | 893 | 0.0723 | 0.125 | −0.694 | 0.0240 | 0.0645 | 0.115 | 1.471 |
| Leverage | 3736 | 0.215 | 0.157 | 0 | 0.0833 | 0.200 | 0.325 | 0.841 |
| Assets | 3788 | 12,423 | 31,367 | 1.038 | 402.0 | 2146 | 9378 | 376,881 |
| BM | 1039 | 0.267 | 0.656 | 0.000436 | 0.0362 | 0.0797 | 0.181 | 4.197 |
| **Panel B: MNCs from Short-Term-Oriented Culture** | | | | | | | | |
| Raw Ret | 10,842 | 14.94 | 78.20 | −99.94 | −25.34 | 5.792 | 37.53 | 1774 |
| α US | 10,761 | 0.0357 | 0.497 | −1.524 | −0.0757 | 0.0224 | 0.121 | 43.87 |
| α UK | 10,761 | 0.0382 | 0.497 | −1.544 | −0.0799 | 0.0295 | 0.130 | 43.90 |
| Abn Ret US | 9634 | 8.600 | 73.54 | −371.8 | −24.60 | −0.120 | 27.15 | 1787 |
| Abn Ret UK | 9634 | 5.993 | 73.65 | −481.9 | −26.53 | −2.385 | 24.52 | 1771 |
| Long-term orient | 10,842 | 37.76 | 12.37 | 6.801 | 25.69 | 36.02 | 51.13 | 51.13 |
| Volatility | 10,842 | 3.044 | 6.939 | 0.123 | 1.742 | 2.429 | 3.500 | 676.0 |
| Haven | 6118 | 0.617 | 0.486 | 0 | 0 | 1 | 1 | 1 |
| Roa | 5276 | 0.0505 | 0.643 | −33.52 | 0.0225 | 0.0766 | 0.137 | 1.138 |
| Leverage | 9437 | 0.226 | 0.370 | 0 | 0.0563 | 0.191 | 0.323 | 25.04 |
| Assets | 10,401 | 6523 | 28,432 | 0.0173 | 101.6 | 659.9 | 3064 | 797,769 |
| BM | 6119 | 0.214 | 0.543 | 0.000436 | 0.0268 | 0.0627 | 0.153 | 4.197 |

Summary statistics for variables in the full sample. Panel A reports for the MNCs from long-term-oriented cultures, and Panel B reports for the MNCs from short-term-oriented cultures. Raw Ret is the annual buy-and-hold return, α US and α UK are α's estimated by regressing daily returns against S&P500 and FTSE100 indices, respectively, Abn Ret US and Abn Ret UK are abnormal returns based on CAPM model in which the market portfolio is proxied by either S&P500 or FTSE100 indices, respectively. *Assets* is in USD millions. *BM* is the book-to-market ratio. *Volatility* is the standard deviation calculated using daily returns for each firm-year combination. *Board size* is the number of people on the board. *Haven* is a dummy equal to 1 when an MNC has at least one tax haven affiliate in the firm structure that it owns more than 50% shares of. *ROA* is the ratio of profit and loss before taxes to total assets, and *Leverage* is the ratio of long-term liability to total assets.

Table 1 shows the means, medians, and distributional information for the firms from countries with more long-term-oriented cultures (Panel A), and from countries with cultures that are more short-term-oriented (Panel B). We can see that the firms from the more long-term-oriented cultures on average have smaller return measures and are larger in size. They are largely comparable to the MNCs from more short-term-oriented cultures in other firm and return characteristics, such as occurrence of extremely negative returns (*crash* and *negskew*), volatility, board size, access to tax havens, and leverage.[6]

### 3.2. Stock Market Returns

In this subsection, we investigate the effect of the WDC on the stock market returns of the affected MNCs. We also split the data by the median of the MNC headquarter country long-term orientation, and conduct the analysis on each sub-sample separately.

In Table 2, we present our results with five stock market return measures on the full sample over 2007–2013. As found in Bilicka et al. (2022b), stock markets respond positively to the WDC, as the difference-in-differences coefficients, *β*, as outlined in Equation (1), are positive and statistically significant across all specifications. Effective tax avoidance can lead to a positive shock to the corporate cash flow, as it allows the firm to pay less tax. Consequently, an anti-tax avoidance rule can cause decreases in the firm value, because

it limits potential tax savings and reduces after-tax income for an investor. However, tax avoidance can also provide means to conceal unfavorable information and questionable corporate practice, and serve as a tool for managerial rent-seeking. From this point of view, the anti-tax avoidance regulation can curb the agency cost by limiting these rent-seeking activities and alleviate information asymmetry. As such, anti-tax avoidance regulation can increase the firm value. The results we present are in line with the second view. Across a broad set of measures, the stock market returns of affected firms have increased after the WDC.

**Table 2.** Annual stock returns: baseline results.

| Dep. Var | (1) Raw Ret | (2) Alpha US | (3) Alpha UK | (4) Abn Ret US | (5) Abn Ret UK |
|---|---|---|---|---|---|
| treated = 1 × post = 1 | 14.59 *** | 0.06 *** | 0.04 *** | 19.48 *** | 17.93 *** |
| | (3.54) | (4.32) | (2.81) | (4.34) | (4.20) |
| MNC level controls | ✓ | ✓ | ✓ | ✓ | ✓ |
| Firm FEs | ✓ | ✓ | ✓ | ✓ | ✓ |
| Year FEs | ✓ | ✓ | ✓ | ✓ | ✓ |
| Observations | 7139 | 7139 | 7139 | 6745 | 6745 |

This table reports the estimated effects of the WDC on MNCs' raw and abnormal returns over the window of 2007–2013. Raw Ret is the annual buy-and-hold return, Alpha US and Alpha UK are $\alpha$'s estimated by regressing daily returns against S&P500 and FTSE100 indices, respectively, Abn Ret US and Abn Ret UK are abnormal returns based on CAPM model in which the market portfolio is proxied by either S&P500 or FTSE100 indices, respectively. *treated* denotes firms that failed the gateway test in 2010; *post* is 1 from 2011 onwards. In all columns, we control for lagged logarithm of total assets, book-to-market ratio, stock return, volatility, and negative skewness against FTSE100. t-statistics in parentheses. *** denotes significance at the 1% level. Standard errors are robust and two-way clustered over MNC group and year.

We also plot the dynamic evolution of the annual buy-and-hold return, *RawRet*, in Figure 1 over a longer time horizon. As documented in Bilicka et al. (2022b), the other return measures follow a similar pattern. We can see that the MNCs affected by WDC outperform firms in the control group after the reform, but only over the first 3-year period.[7] After three years, the return measure largely converges between the control and the treated groups. This short-lived outperformance is consistent with the finding in Bilicka et al. (2022a): the affected MNCs took about three years adjusting their debt and operations, and finished around 2014. Therefore, from then on, the improved information environment due to the WDC does not prevail anymore, as firms are likely to go back to shifting profits using alternative methods. Hence, investors perceive that they may go back to the old practices associated with aggressive tax avoidance behavior. Since the effects of the WDC on both corporate operations and stock market responses only lasted for a limited period, we further investigate whether the different time horizons of investors make a difference for MNCs' stock market behaviors post WDC. We use the long-term orientation dimension of a culture as a proxy to capture the variation in investors' time horizons across different financial markets. Our goal is to examine whether stock markets can recognize the time limit on the WDC's effect found in our empirical analysis and respond accordingly.

Table 3 presents the results. Panel A presents the results for MNCs from countries with cultures that are more long-term-oriented, and Panel B presents the results for MNCs from countries with cultures that are oriented towards the short term. We can see that the positive stock market responses to the WDC documented by Bilicka et al. (2022b) and shown in Table 2 are driven by the firms headquartered in countries with short-term-oriented cultures. The coefficients on the difference-in-differences term, *treated = 1 × post = 1*, are estimated to be positive in Panel B, and significant in four out of five specifications; this is not the case in Panel A. The firms headquartered in countries with more long-term-oriented cultures generally do not respond to the WDC significantly and, looking at the direction of the coefficients, the response tends to be negative. Given that the effectiveness of the WDC is documented to be short-lived, as Figure 1 demonstrates, our finding is consistent with how

Hofstede defines the long-term orientation measure: a country's tendency to focus on the future. When a country where an MNC is headquartered is more inclined to focus on the long-term effect of a shock, investors appear to foresee the transient nature of the WDC's impacts and do not price it into the firm value.

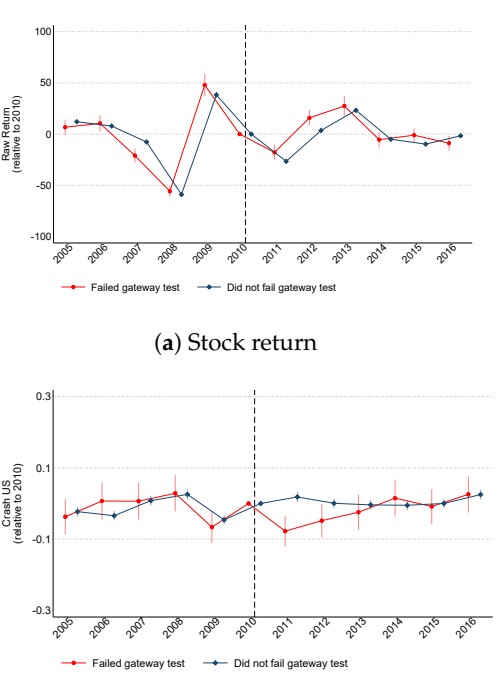

(**a**) Stock return

(**b**) Crash risk

**Figure 1.** Annual stock return and crash risk measures—dynamic plots. Note: These figures plot the average annual buy-and-hold return measure and the crash risk measure from each year benchmarked against year 2010 for both the treated and control groups. We obtain the return measure by regressing *RawRet* on year dummies corresponding to each of the years 2003–2016, and the crash risk measure by regressing *Crash US* on those year dummies. We omit the year 2010; hence, the measures are relative to 2010. Standard errors are robust and two-way clustered over MNC group and year. We plot the 95% confidence interval.

**Table 3.** Annual stock returns: long-term culture vs. short-term culture.

| Dep. Var | (1) Raw Ret | (2) Alpha US | (3) Alpha UK | (4) Abn Ret US | (5) Abn Ret UK |
|---|---|---|---|---|---|
| **Panel A: long-term-oriented cultures** | | | | | |
| treated = 1 × post = 1 | −16.56 | −0.11 | −0.09 | −24.54 | −34.56 ** |
| | (−0.64) | (−1.20) | (−0.99) | (−1.47) | (−2.11) |
| Observations | 861 | 861 | 861 | 825 | 825 |
| R-squared | 0.564 | 0.495 | 0.472 | 0.320 | 0.305 |
| **Panel B: short-term-oriented cultures** | | | | | |
| treated = 1 × post = 1 | 11.33 ** | 0.05 *** | 0.02 | 21.33 *** | 16.30 *** |
| | (2.33) | (2.88) | (0.93) | (3.96) | (3.17) |
| Observations | 5035 | 5035 | 5035 | 4864 | 4864 |
| R-squared | 0.359 | 0.240 | 0.328 | 0.113 | 0.134 |

This table reports the estimated effects of the WDC on MNCs' raw and abnormal returns over the window of 2007–2013. t-statistics in parentheses. ***, ** denote significance at the 1% and 5% level.

### 3.3. Tax Haven Access

In this subsection, we interact the difference-in-differences term from Equation (1) with a dummy variable indicating whether a given MNC has access to any tax haven

subsidiaries. Access to tax havens may indicate that a firm is more "aggressive" in tax avoidance practice (Gumpert et al. 2016; Hines and Rice 1994). Consistent with this remark, Dowd et al. (2017) show that, when shifting profits, firms are more likely to shift more to tax havens. Hence, MNCs that have access to tax havens are more likely to be subject to agency problems due to being more "aggressive" and engaging in more complex financial arrangements that could lead to less information transparency. Bilicka et al. (2022b) find that MNCs with access to tax havens drive the positive stock market responses they find for the affected firms post WDC, consistent with the view that the WDC can mitigate the agency problem for aggressive tax avoiders. However, investors who focus more on the future may hold the view that having access to tax havens suggests that these MNCs can still engage in tax avoidance in the future, which does not help fix the agency problem. Therefore, we expect to see negative signs for the coefficients on the triple interaction terms in the sub-sample of MNCs headquartered in countries with long-term-oriented cultures, and positive signs for the same triple interaction coefficients on the sub-sample of MNCs headquartered in countries with short-term-oriented cultures.

In Table 4, we present results on the effects of the WDC on stock market returns, including the triple interaction with the tax haven dummy. Our goal is to investigate the potentially differential response of the MNCs that have tax haven affiliates in their structure. In Panel A, we show the results using MNCs headquartered in countries with cultures that are more long-term-oriented, and in Panel B using MNCs headquartered in countries with cultures that are oriented towards the short term. The coefficients of the three-way interactions are significantly negative in Panel A, and mostly positive, but insignificant in Panel B. In the long-term-oriented sub-sample, any improvement in the stock market performance in response to the WDC (suggested by the positive coefficient on the term $treated = 1 \times post = 1$ that we estimate), is canceled out by having access to tax havens. These results align with the definition of the Hofstede long-term orientation measure, i.e., a culture's tendency to focus on the future. Since the cultures oriented towards the long term focus on the future, they view the access to tax havens as convenient future platforms for tax avoidance. Hence, the WDC is likely ineffective in curbing agency problems and improving information transparency. The short-term-oriented cultures, instead, are not concerned with the future impacts of the access to tax havens the way their long-term counterparts are.

**Table 4.** Heterogeneity: access to tax havens.

| Dep. Var | (1) Raw Ret | (2) Alpha US | (3) Alpha UK | (4) Abn Ret US | (5) Abn Ret UK |
|---|---|---|---|---|---|
| **Panel A: MNCs from countries with long-term-oriented cultures** | | | | | |
| treated = 1 × post = 1 | 77.24 | 0.21 | 0.25 * | 75.78 * | 40.91 |
| | (1.59) | (1.58) | (1.83) | (1.81) | (0.99) |
| treated = 1 × post = 1× haven | −122.39 *** | −0.40 *** | −0.41 ** | −117.00 *** | −89.17 ** |
| | (−2.59) | (−2.61) | (−2.49) | (−2.85) | (−2.20) |
| Observations | 815 | 815 | 815 | 792 | 816 |
| R-squared | 0.602 | 0.557 | 0.539 | 0.353 | 0.311 |
| **Panel B: MNCs from countries with short-term-oriented cultures** | | | | | |
| treated = 1 × post = 1 | 1.20 | 0.04 | −0.00 | 11.69 | 6.90 |
| | (0.13) | (1.51) | (−0.03) | (1.14) | (0.69) |
| treated = 1 × post = 1× haven | 5.93 | −0.01 | 0.01 | 2.88 | 3.26 |
| | (0.60) | (−0.23) | (0.20) | (0.26) | (0.31) |
| Observations | 4783 | 4783 | 4783 | 4660 | 4660 |
| R-squared | 0.380 | 0.266 | 0.333 | 0.167 | 0.189 |

This table reports the estimated effects of the WDC on MNCs' raw and abnormal returns in connection with an MNC's access to tax haven over the window of 2007–2013. t-statistics in parentheses. ***, **, * denote significance at the 1%, 5%, and 10% level, respectively.

### 3.4. Occurrence of Extremely Negative Returns

To complement our investigation on the differential impacts of the WDC on the stock market returns of the affected MNCs, we examine the occurrence of extremely negative returns as an alternative outcome variable. As shown in Part b of Figure 1, the dynamic effect of the reform on the occurrence of extremely negative returns for the affected firms disappears in 2014. This time limit leads us to conduct a regression analysis on the sub-samples of MNCs divided based on the long-term orientation dimension of their headquarter countries' cultures. We present the results in Table 5. In columns (1) and (2), we report results using dummy variables indicating whether a "crash" type of incident happens in a given year, and in columns (3) and (4) using the negative skewness measures. In columns (1) and (3), we show the likelihood of an extremely negative incidence benchmarked against the US market and in columns (2) and (4) against the UK market. Panel A presents the results for MNCs headquartered in countries with long-term-oriented cultures, while Panel B presents the results for MNCs headquartered in countries with short-term-oriented cultures.

**Table 5.** Crash risk.

| Dep. Var | (1) Crash Risk US | (2) Crash Risk UK | (3) Neg Skew US | (4) Neg Skew UK |
|---|---|---|---|---|
| **Panel A: MNCs from long-term-oriented cultures** | | | | |
| treated = 1 × post = 1 | 0.17 | 0.23 | 0.69 * | 0.65 * |
| | (1.19) | (1.60) | (1.72) | (1.74) |
| post | −0.05 | −0.08 ** | −0.44 *** | −0.48 *** |
| | (−1.57) | (−2.35) | (−5.48) | (−6.27) |
| MNC level controls | ✓ | ✓ | ✓ | ✓ |
| Firm FEs | ✓ | ✓ | ✓ | ✓ |
| Year FEs | ✓ | ✓ | ✓ | ✓ |
| Observations | 3984 | 3984 | 3984 | 3984 |
| R-squared | 0.021 | 0.027 | 0.052 | 0.066 |
| **Panel B: MNCs from short-term-oriented cultures** | | | | |
| treated = 1 × post = 1 | −0.07 ** | −0.03 | −0.20 ** | −0.20 ** |
| | (−2.27) | (−0.82) | (−2.42) | (−2.40) |
| post | 0.34 ** | 0.14 *** | 1.30 *** | 1.57 *** |
| | (2.45) | (3.12) | (3.85) | (2.81) |
| MNC level controls | ✓ | ✓ | ✓ | ✓ |
| Firm FEs | ✓ | ✓ | ✓ | ✓ |
| Year FEs | ✓ | ✓ | ✓ | ✓ |
| Observations | 5035 | 5035 | 5035 | 5035 |
| R-squared | 0.023 | 0.029 | 0.069 | 0.086 |

The dependent variable is one of the proxies for extremely negative returns. $CrashUK(US)$ is based on whether the firm-specific weekly return has dropped 3.2 standard deviation below the annual average in a particular year estimated against FTSE100(S&P500). $NegSkewUK(US)$ is the negative skewness of the weekly return in a specific year estimated against FTSE100(S&P500). In all columns, we control for lagged logarithm of total assets, book-to-market ratio, stock return, volatility, leverage, and ROA. In columns 1 and 3, we control for lagged UK crash risk and in columns 2 and 4 we control for lagged US crash risk. t-statistics in parentheses. ***, **, * denote significance at the 1%, 5%, and 10% level, respectively. Two-way firm and year clustered standard errors in parentheses.

Again, MNCs headquartered in countries with cultures with different levels of long-term orientation behave differently. Previously, Kim et al. (2011) documented the association of the occurrence of extremely negative returns with tax avoidance activities, and Bilicka et al. (2022b) showed that the affected firms experience less occurrence of extremely negative returns post WDC . We only find significant empirical results consistent with these previous findings in the sample of MNCs headquartered in countries with short-term-oriented cultures. The WDC significantly reduced the crash risk the affected MNCs were subject to, but only for those MNCs headquartered in countries with short-term-oriented cultures.

## 4. Conclusions

In this paper, we examine the differential stock market reactions to a change in anti-tax avoidance regulations, depending on the national culture of the country where the MNC is headquartered. Using the introduction of the Worldwide Debt Cap (WDC) reform in 2010 in the UK as a quasi-natural experiment, we examine the role played by the long-term orientation dimension of national cultures in the impacts of the WDC on stock market behaviors of the MNCs affected by the reform.

Consistent with the literature, we find that MNCs affected by the reform were associated with higher subsequent stock market returns than their unaffected counterparts after the reform; however, the effect disappeared after three years. This is consistent with the evidence that multinational firms eventually moved debt and real operations away from the UK in response to the reform, thereby diminishing the effects of this anti-tax avoidance regulation. In line with the transient nature of the WDC's impacts on the affected MNCs' stock market performance, we find that investors from countries with cultures oriented towards the long term appear to recognize that these impacts can be short-lived. When dividing our sample by the median of the Hofstede long-term orientation dimension of the national culture, we find that the positive stock market response to the WDC and the decrease in occurrence of extremely negative returns are primarily driven by MNCs headquartered in countries with short-term-oriented cultures. Cultures with different terms of orientation also view the role of tax havens in this episode differently. Investors from countries with long-term-oriented cultures focus more on the future, thereby viewing the access to tax havens as a future potential to circumvent the anti-tax avoidance regulation. Investors from countries with short-term-oriented cultures, by contrast, do not exhibit such a future-oriented view.

From a policy perspective, we provide evidence on the extent of anti-tax avoidance regulations' effectiveness, focusing on the heterogeneous impact of these regulations, which varies depending on differences in countries' long-term orientation. We document that the long-term orientation dimension of a country's culture plays an important role in explaining the cross-country variation in the stock market responses to the WDC. This heterogeneous effect that depends on the forward-looking perspective of the investors in a given country is important for policy makers as they consider implementing similar anti-tax avoidance regulations in the future. As of January 2024, many countries have started implementing a Global Minimum Tax to curb the global tax avoidance practices of MNCs.[8] The evidence we provide in this paper highlights that regulations that limit profit shifting affect the stock returns of MNCs across countries differentially, and these effects should be taken into account by policy makers as they implement these new regulations targeting MNCs. While our goal in this paper is to make inferences about the general effectiveness of anti-tax avoidance regulations, we acknowledge that this study focuses on a single reform. Conducting a comparative analysis involving other anti-tax avoidance regulations could offer additional insights on the effectiveness of such regulations in other countries and further inform policy makers. This is beyond the scope of the current paper but is a promising avenue for further research.

Finally, and more broadly, our findings not only advance the field of international finance, but also have implications for how long-term beliefs affect asset prices in a global setting. This addresses a key question in asset pricing research: the significance of incorporating investors' long-run return expectations into asset pricing (Brunnermeier et al. 2020).

**Author Contributions:** Conceptualization, K.B., D.C.-S. and Y.Q.; methodology, D.C.-S.; software, D.C.-S.; validation, K.B. and D.C.-S.; formal analysis, D.C.-S.; investigation, K.B. and D.C.-S.; resources, K.B. and Y.Q.; data curation, K.B. and Y.Q.; writing—original draft preparation, D.C.-S.; writing—review and editing, K.B. and D.C.-S.; visualization, K.B. All authors have read and agreed to the published version of the manuscript.

**Funding:** This research received no external funding.

**Data Availability Statement:** Data can be provided upon request

**Conflicts of Interest:** The authors declare no conflict of interest.

## Notes

[1]    For instance, Beugelsdijk et al. (2021) present an overview of the business studies that have applied the Hofstede cultural measures since 2006.

[2]    See https://geerthofstede.com/research-and-vsm/dimension-data-matrix/, accessed on 1 April 2022.

[3]    *Raw Ret* is obtained by compounding monthly returns within each year. *Alpha UK(US)* is the CAPM alpha against FTSE100 (S&P500). *Abn Ret UK(US)* is the CAPM adjusted abnormal return against FTSE100 (S&P500) using $\beta$ estimated from monthly returns over the previous 36 months.

[4]    A "crash week" is defined as a week when the firm-specific weekly return drops 3.2 standard deviations below their annual mean value, with 3.2 chosen to generate a frequency of 0.1% in the normal distribution. The firm specific weekly return is defined as $W_{i\tau} = ln(1 + \epsilon_{i\tau})$, where $\epsilon_{i,\tau}$ is obtained from the regression of weekly return $r_i$ on weekly market index returns $r_m$ and its leads and lags.

[5]    The negative skewness (*Neg Skew*) are computed as follows,

$$Neg\,Skew_{it} = -[n(n-1)^{3/2} \sum W_{i\tau}^3]/[(n-1)(n-2)(\sum W_{i\tau}^2)^{3/2}]$$

where $W_{i\tau}$ is the firm-specific weekly return for firm $i$ in week $\tau$, and n is number of available weekly returns in year $t$. The negative skewness measures are also estimated against both FTSE100 and S&P500.

[6]    Bilicka et al. (2022b) show that before the WDC, the MNCs in the control group were in general significantly larger, but did not have significantly different ROAs and leverage. There is also no significant difference in the stock market performance between control and treatment groups before the WDC, except for alphas in the US. Treated group is defined as firms with gateway test ratios above 75% in 2010, therefore has higher gateway test ratios, as expected.

[7]    The graphs in Figure 1 also suggest that there was no significant difference in stock market performance between treated and control groups before WDC. Standard errors of coefficient estimates overlap in most years prior to the reform. Further, the lack of differential trend evolution between treated and control group firms in years immediately after the financial crisis, but before the WDC reform, suggests that the financial crisis affected all firms in our sample to a similar extent.

[8]    See https://www.oecd.org/tax/beps/summary-economic-impact-assessment-global-minimum-tax-january-2024.pdf, accessed on 1 February 2024.

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
