# Peer review of "Long-Term Orientation and Tax Avoidance Regulations"

_jrfm, doi:10.3390/jrfm17030102_

Round 1

Reviewer 1 Report

Comments and Suggestions for Authors

The importance of the paper is not clear. Some statements are not funded, for example why the access to tax havens or the quality of government is responsible for the stock returns of MNCs? The terms used in the introduction are not explained well. For example, what does it mean "long-term oriented culture" or "different view on tax haven in association with WDC [World Debt Cap]"? Can a view be associated with the regulation? The effect of short-term oriented culture for returns is positive but for long-term oriented culture they "do not experience significant reductions in extreme negative returns". What does it mean? Are they positive, less negative, or insignificant? The paragraph about Brunnermeier is redundant.

There are a lot of self-citations at the beginning of the paper (Bilicka 2022). The base concept comes from old literature Hofstede et al (1991).

The method is not sufficiently explained. What is the role of abnormal rate of return? Why these variables and methods are used? The crucial notions are neither presented nor discussed but instead, we read "We obtain the culture measure from Hofstede’s website and merge it with the corporate financial and stock return data." So what is measured and what is the the purpose of this study?

The interpretations based on Figure 1 are not convincing as we can observe many periods with lines below or above each other. Extending the period before and after the change could affect the results (e.g. to 5 years before and after the change).

Table 3 In three cases the result is significant for short-term culture but not significant for long-term culture but in two cases (Alpha U and Abn Ret UK) the results are insignificant or significant in both cases. Is this enough to say that these observations are consistent with the predictions? So why do such discrepancies occur? Aren't they the result of the variables used to describe the returns?

The result that "long-term oriented cultures focus more on the future, while short-term oriented cultures focus more on the present" sounds trivial. In fact, we know only that access to tax havens affects MNCs differently from different cultures. The insignificance of parameters in the model of short-term firms having access to tax haven tells us nothing about the sources of return improvement.

WDC effect is transitory but the conclusion suggests that it is effective (at least for long-term culture MNCs).

Comments on the Quality of English Language

The explanations in the introduction are not clear. The sentences include too many threads. The structure of the paragraphs is chaotic, particularly in the introduction.

Reviewer 2 Report

Comments and Suggestions for Authors

Dear Authors

Thank you for giving me the opportunity to review your paper, 'Long-Term Orientation and Tax Avoidance Regulations.'

The article 'Long-term orientation and tax avoidance regulations' proposes the exploration of the relationship between the cultural orientation of the country and the reaction of the MNC stock market to tax avoidance regulations.

Even though this topic is of high interest for researchers and professionals in the field and that the current paper uses a consistent quantitative approach to demonstrate the findings relevance, I would suggest some improvements that might increase the scientific quality of your work:

1.     Extend the literature review analysis to other recent contributions to the topic that would increase the relevance of the paper (an extended geographical area: region, global etc)

2.     Provide more details on how your findings are supporting (or not) the conclusions of the relevant literature and, eventually, how your research is filling the gap. It is also important to address the global relevance of your findings to other countries, regions, stock markets etc.

3.     Extend your conclusions with the limitations of your paper and with the future continuation that you foresee.

Best regards

Reviewer 3 Report

Comments and Suggestions for Authors

Respected Authors,

The paper presents an interesting approach to an important subject. Overall, this was an interesting and valuable paper that was a pleasure to read.

However, I have minor comments that can be easily resolved.

1-    First, looking for the data, authors considered an annual database ranging from 2005 to 2016. Why you had not considered an updated data base till 2022.

2-    Authors used the difference-in-differences (DID) approach without explaining this method. I think, that as the authors give more explanation on variables used in this work, they must give an explanation to the methodology concerning the difference-in-differences approach.

3-    How did authors ensure robustness? I think that a robustness analysis is required to check the validity of the adopted methods.

4-    The practical implications of the findings need to be discussed further. They are just very briefly and shallowly highlighted. The conclusion needs to be extended to include more implications for policymakers.

Kind regards.  

Round 2

Reviewer 1 Report

Comments and Suggestions for Authors

Dear Authors,

The revised version is better drafted and more relevenant.

Regards,

Author Response

Thank you for your feedback.

Reviewer 3 Report

Comments and Suggestions for Authors

Dear Author,

Thank you for your efforts to make the required corrections and spend time to improve the quality of the manuscript.

Looking for this new version we can notice that authors take into account all mentioned comments by reviewers and clarified the required corrections. Although database is old, results can be generalized. Authors clarified the methods employed in this research and clarified the empirical results. The discussion of the results in this updated manuscript are clearly presented.

I think that minor corrections in the form must be considered in this updated manuscript. First, references must be written in an ordered manner. Second, quality of English language can be more improved.

In conclusion, paper can be accepted for publication in the Journal of Financial Risk and Financial Management after the correction of these minor mentioned comments.

Sincerely Yours,

Comments on the Quality of English Language

The quality of English language can be more improved.

Author Response

Dear Reviewer,

Thank you for your suggestions. In response to your feedback, we have made the following changes in the current manuscript:

  1. We have adjusted the reference style to present the authors of the cited studies in a more organized manner.

  2. The English language has been edited as per your recommendation.

We appreciate your time and effort in reviewing our work.

________________________

End of Response.